# Oncolytic Virotherapy Treatment of Breast Cancer: Barriers and Recent Advances

**DOI:** 10.3390/v13061128

**Published:** 2021-06-11

**Authors:** Amy Kwan, Natalie Winder, Munitta Muthana

**Affiliations:** Department of Oncology and Metabolism, University of Sheffield Medical School, Beech Hill Road, Sheffield S10 2RX, UK; amy.kwan@sheffield.ac.uk (A.K.); njwinder1@sheffield.ac.uk (N.W.)

**Keywords:** oncolytic virus, breast cancer, immunotherapy, immunogenic cell death, menopause

## Abstract

Oncolytic virotherapy (OV) is an emerging class of immunotherapeutic drugs. Their mechanism of action is two-fold: direct cell lysis and unmasking of the cancer through immunogenic cell death, which allows the immune system to recognize and eradicate tumours. Breast cancer is the most common cancer in women and is challenging to treat with immunotherapy modalities because it is classically an immunogenically “cold” tumour type. This provides an attractive niche for OV, given viruses have been shown to turn “cold” tumours “hot,” thereby opening a plethora of treatment opportunities. There has been a number of pre-clinical attempts to explore the use of OV in breast cancer; however, these have not led to any meaningful clinical trials. This review considers both the potential and the barriers to OV in breast cancer, namely, the limitations of monotherapy and the scope for combination therapy, improving viral delivery and challenges specific to the breast cancer population (e.g., tumour subtype, menopausal status, age).

## 1. Introduction

Neoplastic disease accounts for one in six deaths globally, with cancer being the second leading cause of death worldwide. The most frequently occurring subtype of malignant cancer found in woman globally is breast cancer, with statistics showing roughly 1000 women dying per month in the UK [1]. Even though breast cancer is localized, it can be described as a heterogeneous disease, exhibiting multiple phenotypic variations [2]. These genetic variations, combined with other factors including tumour size, grade, and morphology alongside hormone receptor expression, are also used for diagnostic purposes to stratify patients’ prognosis and treatment regime [3]. The most abundant subtype of breast metastases stems from adenocarcinomas, accounting for 95% of all invasive breast cancer cases recorded. From this population, 55% will present with invasive ductal carcinoma (IDC), which is characterised by the uncontrolled neoplastic proliferation of epithelial cells, which are localized to the ducts or lobules of the mammary gland [3]. Of these IDC patients, 15% will develop a triple negative breast cancer (TNBC) status (Cancer Research UK, 2017). TNBC is a highly metastatic disease that has been known to spread to distant organs such as the brain, bone, and the lungs. The inability to remove residual cells from the primary site after the initial treatment is ceased increases the risk of multi-drug resistance (MDR) due to genetic heterogeneity, enabling tumour progression [4]. Moreover, the genetic variations of this cancer subtype leave the individuals negative for the hormone receptors human epidermal growth factor (HER-2), oestrogen (ER), and progesterone, resulting in a lack of response to traditional hormone therapies and elevating the problems associated with breast cancer therapies [3]. The current therapy options available for the treatment of aggressive breast cancers consists fundamentally of hormone therapies, human epidermal growth factor receptor 2 (HER2)+ receptor targets, and chemotherapies, used sometimes in combination with immunotherapies. Unfortunately, evidence still suggests that some patients die as a result of harsh treatments, with harmful side effects overshadowing patient benefit [4]. Such evidence highlights the desperate need for research to uncover new, alternative therapies in the fight against breast cancer.

In the new era of scientific innovation, genetic engineering has led the way in the field of cancer research, with oncolytic viruses sparking new interest. Multiple viruses have been utilized within the field of oncology, including adenovirus, reovirus, measles, herpes simplex, Newcastle disease, vaccinia, and Myxoma viruses. Oncolytic viruses are either inherently tumour specific (e.g., Myxoma virus) and display a natural tropism towards tumours cells or modified to enhance tumour specificity. The relationship between virus and tumour cells was first discovered in the early 1800s, when patients with leukaemia and lymphoma exhibited tumour regression after contracting the measles virus [5]. Since then, genetic engineering has enabled researchers to enhance viral tumour specificity through the deletion or insertion of essential genes within their genomic structure. For example, the most recent virotherapy to be approved by the Food and Drug Administration (FDA) and the European Medicines Agency was a herpes simplex virus (HSV-1) known as Talimogene laherparepvec (T-Vec). The T-Vec virus was attenuated to express high levels of granulocyte macrophage colony-stimulating factor, which is an essential cytokine for the production and stimulation of new infection-fighting white blood cells [6]. It was first used as a single therapy for the treatment of aggressive melanoma. However, research has since looked at its effects in combination with immunotherapies. Sun et al. (2018) [7] recently investigated the combination effects of a checkpoint inhibitor and T-Vec in unresectable stage III-IV melanoma. The study was conducted on a small number of patients (n = 10) with over half (n = 6) experiencing complete response after intratumoural injection of T-Vec, with an additional two patients seeing a response in not just the primary lesion but off-target lesions as well. Follow-up blood work showed that in patients exhibiting a response, CD4+ and CD8+ T cells were elevated seven -months post treatment, suggesting good synergy between the two therapies and a possible treatment for enhancing immune activation in cancer [7]. Nevertheless, the sobering fact is that at present such immunotherapies work in a limited patient population with a given cancer type, and in some types of cancer they have little or no effect. Breast cancers have historically been amongst the hardest to treat and so far, immunotherapies including checkpoint inhibitors have resulted in little success in clinical trials [8,9]. This highlights the need for new combinational therapies that help target cancers that are specifically immunosuppressed, using checkpoint inhibitors.

The method by which oncolytic viruses specifically kill tumour cells is still under investigation; however, it is thought that the main mode of cancer killing is through direct oncolysis and increased tumour immunogenicity culminating in neoplastic cell elimination. The ability for oncolytic viruses to generate multiple daughter virions upon infecting one tumour cell is one of the more attractive qualities of using virotherapies. This “self-implication” property enhances rapid tumour lysis and increases the possibility of single or lower dosing regimens for patients. The fact that cancerous cells deviate from their original homeostatic signals enables great oncolytic viral sensitivity [10]. Besides oncolysis, immune activation is another key aspect to tumour eradication and is arguably the most important. Therefore, virotherapy can switch a previously immunogenically “cold” tumour into a “hot” one through the exposure of tumour-associated antigens (TAA) from the primary cancer to the circulatory system, thus eliciting an anti-tumour response from the immune system [11]. TAA stimulation of the immune system via OV offers the potential for long-lasting tumour immunisation from re-occurring cancers by generating tumour-specific T cells [10]. This is a major advantage and has potential to make tumours more responsive to immunotherapies that would ordinarily not work.

Although vaccines of viral components, in the form of gene therapy, can often be thought of as an OV, this review focuses on the use of live, replicating viruses and their potential role in breast cancer treatment.

## 2. Preclinical Evidence of OV in Breast Cancer

Breast cancer is a heterogeneous disease where the majority of tumours are immunologically “cold,” and therefore the use of immunotherapeutics within breast cancer appeals to be less compelling than those of melanoma or lung cancer. However, OV has been shown to modulate the tumour microenvironment (TME), causing an increase in pro-inflammatory cells (e.g., cytotoxic T cells, macrophages) that could potentially turn a “cold” tumour into a “hot” one (Figure 1). In breast cancer, there is a number of preclinical studies using oncolytic virotherapy as well as a few early-phase clinical studies with promising early markers of response [12,13,14,15,16,17,18]. This was reviewed recently by O’Bryan and Mathis [19], Martini [9], and Chaurasiya and Fong [20]. 

Some cancers such as melanoma and lung cancer demonstrate high response rates to immune checkpoint inhibitors and are commonly referred to as “hot tumours.” These are in sharp contrast to tumours with low immune infiltrates, called “cold tumours” or non-T-cell-inflamed cancers, such as breast cancer. OV has the potential to make “cold” tumours “hot” by reprogramming the TME in patients whose tumours are cold. The virus can inflame the tumour, attract immune antigen-presenting cells, and recruit T cells that can kill tumour cells. 

Oncolytic viruses can be broadly categorized into DNA and RNA viruses. The DNA viruses most commonly investigated in breast cancer are adenovirus, HSV, and varicella. Oncolytic adenoviruses may be easily modified by the insertion of tumour-specific promoters to ensure replication occurs only with tumour cells. This can be used to improve breast cancer targeting. The receptor by which most adenoviruses enter the cell is the human Coxsackie receptor (hCAR); however, this is found in low numbers on breast cancer cells. Therefore, a study recently attempted to improve breast cancer targeting by the placement of an Ad3 component into the Ad5 backbone, which allows the adenovirus to enter cells through non-hCAR routes. Furthermore, an oncolytic adenovirus coding for GM-CSF was engineered and shown to lead to clinical stable disease in 8/12 patients who were advanced refractory to conventional treatments, with resolution of pleural fluid and ascites in some patients [21]. No breast cancer patients were included in this small study, and therefore we are unable to speculate whether this response would have been seen for this group.

Additionally, promoters may be inserted to enhance viral efficacy. Insertion of a promoter to the transcriptional factor E2F-1 and IL15 showed decreased tumour growth in both in vitro and in vivo models of TNBC using MD1-MB-231 cells inoculated in nude mice. Furthermore, a dual cancer-specific oncolytic adenovirus Ad-Apoptin-hTERTp-E1A (Ad-VT) with the apoptin and hTERT promoter was constructed and showed improved cytotoxicity compared to non-modified virus in MCF-7 breast cancer spheroids grown in vitro [22]. 

Herpes simplex type 1 (HSV-1) is a double-stranded DNA virus that fuses to the plasma membrane of host cells. To limit viral replication to cancer cells, the ICP34.5 gene is deleted, which results in an inability to replicate within neurons. Additional modifications can be made to the HSV-1 envelope to allow increased specificity to breast cancer cells. For example, the HSV G47(delta) has additional mutations in ICP6 and (alpha)47. This variation has demonstrated in vitro and in vivo effectiveness. Of particular interest is the potential for the use of this virus in tamoxifen-resistant breast cancer [23]. 

A naturally occurring variant of HSV, HF10, has been explored in breast cancer in the preclinical and clinical settings, with patients showing decreased tumour size and increased CD8+ T-cell infiltration in the TME. In one such study, patients were given a dose of HF10 into one metastatic nodule and a dose of placebo into another [24]. The authors commented that the nodule treated with HF10 had signs of immune activation with increases in infiltration of CD8+ T cells and increased apoptotic markers. In our work with HSV1716, we have found systemic immune responses to OV treatment in a number of immunocompetent murine models of TNBC. This did, however, appear to be more marked when treatment was delivered systemically as opposed to locoregionally [25]. 

A number of RNA viruses have also been studied for use in breast cancer. Reovirus is a double-stranded RNA virus that is naturally non-pathogenic to humans and therefore used in an unmodified form. In particular, the type 3 Dearing reovirus strain is naturally oncolytic, and several studies have been performed that used it against breast cancer [26,27]. Due to early promise, intralesional administration of reovirus has undergone a phase 1 trial in advanced cancers, of which three patients had advanced breast cancer. As this revealed tolerable side effects, further studies including breast cancer patients have involved combinations of reovirus with other therapies, including phase 1 combinations with chemotherapies (docetaxel [28] and gemcitabine [29]), and in vivo studies with immunotherapies (PD-1 inhibitor) [30]. The combination of OV and immunotherapies has shown most promise and is discussed later. 

## 3. Barriers to OV

Although OV appears promising, there are several limitations. These can be broadly categorized into efficacy as monotherapy and challenges with delivery (Figure 2). 

The approach to many novel therapies has been to ascertain safety, efficacy, and biological mechanisms of action through rigorous testing of oncolytic viruses as monotherapies. However, the actual efficacy of OV on its own is limited, as is the case with more established immunotherapies such as checkpoint blockade. Figure 2 outlines the challenges that need to be overcome and potential strategies for improved delivery of oncolytic viruses.

This includes neutralizing oncolytic viruses in the bloodstream, sequestration of oncolytic viruses in the liver/spleen leading to decreased viral titre arriving at the site of the tumour, imbalance within the TME leading to poor tumour penetration, and suboptimal TAA production (which may be enhanced through a combination of OV and conventional treatments).

Additionally, cancer therapies, including virotherapy, carry their own risk and the way in which these treatments are administered can greatly reduce these risks. Most oncolytic viruses, especially those undergoing clinical trials, are predominantly administered through intratumoural (IT) injection. This helps overcome one of the main obstacles associated with intravenous administration, such as viral neutralisation and sequestration. Many studies, including those by Andtbacka et al. [31] and Cripe et al. [32], utilised IT delivery to establish large viral titres within the TME and generate an anti-tumour response.

The following sections will address the developments that have been made to overcome these challenges and ascertain whether breast cancer itself may pose a barrier to OV therapy.

## 4. Combination with Chemotherapy

Patient stratification is not only used for determining patient prognosis but also for identifying the best treatment regime. Before determining the best course of action for the patient, decisions need to be made on whether the breast cancer can be eradicated with surgery or mastectomy, and whether chemotherapy should be given prior to the surgery (neoadjuvant therapy) or after (adjuvant treatment). Personalising chemotherapy treatments with breast cancer subtypes is essential for patient survival, with many chemotherapies targeting different phenotypic variations within the cancer. Such chemotherapies are used for the management of excessive cell proliferation by targeting DNA repair mechanisms with platinum, causing DNA damage with anthracycline agents and inhibiting P53 synthesis with taxanes [33]. However, many of the current therapies used within the clinic today lack the ability to target tumour cells specifically and therefore are administered in large doses in order to eliminate residual breast cancer cells. Unfortunately, this approach usually leaves the patient with poor quality of life due to harmful off-target side effects resulting from a lack of treatment specificity. Therefore, combinational therapies comprising OV and lower doses of chemotherapy could be a potential alternative treatment option for breast cancer patients.

At present, there are very few chemo-virotherapeutics making headway in breast cancer clinical studies. However, promise has been seen in other cancer types. In advanced phase II clinical studies, Karapanagiotou et al. [34] investigated the synergy of a reovirus type 3 Dearing (RT3D) virus, which is attenuated to specifically replicate in RAS-transformed cells and carboplatin/paclitaxel in patients with relapsed or metastatic solid tumours. Previous studies conducted by the same group determined optimal dosing regimens prior to the phase II trial mentioned above. Patients received a combinational therapy of intravenously administered chemovirotherapy (virus ((RT3D virus) given over five days, whilst chemotherapy (carboplatin/paclitaxel) was given three times a week. Tumour response was evaluated alongside any evidence of an anti-tumour response (n = 19). The results demonstrated a complete response in one patient, whereas eight went on to experience a partial response to treatment. In the remaining patients, nine had a stable disease state and eight had disease progression. The safety profile of the study showed that patients tolerated treatment extremely well, with minimal to no known adverse effects, and was considered a good treatment option for patients with head and neck cancer [34]. Furthermore, a randomized study of pelareorep and paclitaxel in advanced breast cancer did not show a difference in progression-free survival, but rather a significantly longer OS with the combination [35]. So far, most clinical investigations have focused on the treatment of solid tumours in easily accessible areas, where the oncolytic virus can be injected directly into the tumour. This leaves breast cancer on the outskirts of clinical research. However, promising pre-clinical investigations are currently being conducted to help overcome this. Berry et al. [36] developed a doxorubicin conjugation reovirus (re-dox) for controlled drug release whilst simultaneously enabling viral lysis of tumour cells for the treatment of TNBC. The group used a hetero-bifunctional crosslinker (succinimidyl 4-(N-maleimidomethyl)cyclohexane-1-carboxylate), which enables covalent bonds to form between the doxorubicin and virus, and enables the simultaneous release of viral particles and doxorubicin within the tumour after intratumoural administration. The results demonstrated in vitro show that the combination therapy increased mRNA expression of innate immune activation markers, including interferon—IFNL1, IFNB1, and IFNG in MDA-MB-231 cells—whereas treatment with re-dox still retained its infection and DNA-damaging abilities. Re-dox also significantly reduced tumour burden in mice with TNBC (4T1 model) implanted in the hind flank, resulting in a reduction in metastatic disease, predominantly within the lungs [36]. In addition, Bourgeois-Daigneault et al. looked at the combination of oncolytic rhabdovirus Maraba-MG1 and paclitaxel for the treatment of murine TNBC in two established cell lines, 4T1 and E0771, in vitro and in vivo [12]. In vitro results demonstrated synergistic behaviour between the two therapies, with elevated viral propagation in 4T1 tumour cells. Further investigation also evidenced that paclitaxel does not affect the infection or replicative ability of the oncolytic virus. Mice were implanted with 4T1 an EMT6 cells in the second left mammary fat pad for in vivo experiments with treatments administered via the intraperitoneal or intratumoural route. The results showed significant tumour killing compared to controls when combination treatments were used over individual treatment [12]. Data such as these demonstrate the potential for OV in combination with chemotherapies as a future treatment option for breast cancer. However, other combinations need to be investigated to identify which options are most effective for breast cancer, including radiotherapy and immunotherapy.

## 5. Combination with Radiotherapy

Radiotherapy is a modality of primary treatment in 50% of cancers. It is an effective treatment on its own and causes cell death through DNA damage that may not be repairable in areas of malignancy. In a review by O’Cathail et al., they described how the combination of an oncolytic adenovirus and radiotherapy can be used to sensitise tumours to radiotherapy treatment. They postulate that this is due to the adenovirus preventing DNA repair following DNA damaging radiotherapy treatment leading to cancer cell death [37].

However, to date, there is little clinical data about the effectiveness of such treatment and no such data in relation to breast cancer. T-Vec, as mentioned earlier, is an HSV-derived virus that has been approved for use in melanoma. Within this population of patients, the combination of radiotherapy and T-Vec has been explored in preclinical animal models, with promising results of synergistic effects [38]. In addition, a phase II trial of intratumorally administered T-Vec in combination with hypofractionated radiotherapy in melanoma and other tumours commenced in 2016, with results expected later this year.

Other promising trials in active recruitment include the Chemoradiation with Enadenotucirev as a radiosensitiser in locally Advanced Rectal cancer (CEDAR) trial, which is a dual-endpoint, dose-escalation phase I trial of an intravenously administered adenovirus, enadenotucirev, in combination with radiotherapy in colorectal cancer. This route of administration may be advantageous in the cancer population, as it allowed multiple areas to be sensitized to radiotherapy following a single dose of OV treatment [39].

The mechanism of this synergy is not fully clear. Many studies speculate on the combination of DNA damage secondary to radiotherapy and the viral properties of preventing DNA repair. However, some groups have found that the addition of radiotherapy to OV treatment results in enhanced viral replication, viral yield, and viral release. In this study of adenovirus dl520, the addition of radiation inhibited the growth of subcutaneous U373 glioblastoma tumours in a xenograft mouse model through an increase in YB-1, a protein required for viral replication [40]. Given the widespread use of radiotherapy in curative and metastatic breast cancer, the potential for a synergistic therapy, and the paucity of available studies, it would seem pertinent to assess this combination of OV and radiotherapy in breast cancer.

## 6. Combination with Immunotherapies

In a recent publication, we showed that the breast TME becomes primed for immunogenic killing through the use of oncolytic viruses. In particular, the phenotype of tumour-associated macrophages (TAMs) becomes re-educated from a tumour-promoting M2 subtype to a more inflammatory and cytotoxic M1 phenotype. These TAMs also enhance the recruitment of cytotoxic CD8+ T cells that directly eradicate cancer cells. However, we also showed that although OV may increase the number and activation of cytotoxic immune cells within the TME (CD8+, M1-like macrophages), they also upregulate the expression of PD-L1 within the tumour, which causes an arrest to this immunological killing [25]. This led to speculations that OV therapy in combination with PD-L1 or PD1 inhibitors may release this break, allowing the immune system to respond more actively to OV.

Bourgeois-Daigneault et al. described the use of a Maraba virus prior to the removal of breast tumours in immunocompetent murine models [12]. Here, they administered a course of OV treatment 7 days prior to mammary tumour resection, followed by an adjuvant course of PD1 inhibitor treatment. They demonstrated that the use of virotherapy prior to surgery allowed for the sensitization to immune checkpoint therapy given adjuvantly and that on rechallenge, the immunological effects were long lasting [41]. This novel, neoadjuvant approach to treatment is appealing to clinical trial design, where the degree of response to neoadjuvant treatment can guide whether further adjuvant treatment is required.

Similarly, Mostafa et al. demonstrated that the oncolytic effect of reovirus can be enhanced with the addition of a PD-1 inhibitor in an EMT6 immunocompetent murine model of breast cancer, causing a reduction in tumour growth in comparison to monotherapy [30]. Furthermore, this is likely due to the positive cytotoxic changes in the immune TME, including an increase in CD8+ cells and a decrease in CD4+ T cells. A synergistic effect of OV and PDL1 inhibitor was seen in the study by Chaurasiya et al., where a pox virus, CF33-hNIS-ΔF14, was used in combination with an immune checkpoint inhibitor (anti-PD-L1) using the EO771 murine model of breast cancer. Here they observed no significant change in tumour size with treatments as monotherapy, but the combination of both treatments lead to a 50% survival rate for animals at 100 days post treatment. They also noted the change in the TME, with an increase in pro-inflammatory cytokines and activated CD8+ T cells [42].

Given the promising preclinical data (Table 1), there are a few ongoing clinical trials combining oncolytic viruses with a checkpoint inhibitor in breast cancer. Viral groups of investigation at present include HSV, vaccinia, and reovirus, with the likelihood that other viral groups will be included as data mature.

## 7. Overcoming Barriers of Intravenous Delivery OV

The haemodynamics of the TME is a key aspect associated with drug delivery, with most therapies relying on the “leaky” vasculature of the tumour for drug uptake. Major efforts have been exerted on the enhanced permeability and retention (EPR) effect, also referred to as the “royal gate” of drug delivery, since its importance was first highlighted in the late 1980s [45]. The combination of vascular fenestration and collapse, the heterogeneity of the basement membrane, the dense coverage of pericytes, and the lack of lymphatic blood vessel formation all contribute to elevated interstitial fluid pressure (IFP) within tumours. Increased IFP leads to interstitial hypertension within the TME, which restricts the network of connective transport systems available for OV perfusion and extravasation, diminishes the EPR effect [46], and provides a challenge for intravenous (IV) delivery. The importance of the EPR effect for efficient drug delivery was demonstrated following the systemic delivery of oncolytic viruses in heterogeneous intratumoural (IT) perfusion states within the TME [47]. Within the study, animals bearing myeloma tumour cells were administered with oncolytic VSV whilst undergoing physical exertion. Exercise is known to decrease splanchnic circulation, preventing viral sequestration and enhancing the rate of blood flow, which increases IT perfusion pressure. The results indicated significant amplification in the amount of “infection centres” and greater homogenous infection distribution throughout the tumours. This correlated with greater overall survival [47]. However, exercise is not a feasible option for all patients, making IT a better delivery option for oncolytic viruses. The successful delivery of therapies in breast cancer remains a major obstacle in clinic, with the TME arguably being the main candidate that drives drug resistance. Shee et al. [48] recently published a study stating that factors secreted by the TME, mainly the angiogenic cytokine fibroblast growth factor 2 (FGF2), which promotes tumour progression via irregular vascular formation, was highly overexpressed within ER+ breast cancer cells after in vivo experiments of immunocompromised mice bearing MCF-7 xenografts. The results demonstrated that FGF2 modulated resistance to fulvestrant and other PI3K–mTOR pathway inhibitors in anti-oestrogen-resistant ER+ breast cancer [48]. This provides a possible therapeutic targeting strategy via FGF2 mediated pathways, which could remove the TME as an uptake barrier for oncolytic viruses, and highlights a need for personalised treatment in relation to individual patient TME status.

However, IT delivery poses challenges of its own, including being a limited delivery option for inaccessible tumours or small metastatic lesions that cannot be reached with a needle. In cases where IT delivery is not possible, IV delivery is the next best option. Therefore, “camouflaging” oncolytic viruses is of utmost importance to increase therapeutic efficacy in patients, particularly as many of us have already been exposed to these viruses and will therefore carry pre-existing neutralising antibodies (NAb) that will prevent the virus from entering the tumour. The encapsulation of oncolytic viruses through both synthetic and biological agents such as immune cells, copolymers, nanoparticles, and biodegradable materials is the main contender for viral “cloaking” [49,50,51,52]. Research carried out by Muthana et al., (2011) demonstrated the potential of cell carriers, namely, “macrophages,” as biological protectors of a prostate-specific adenovirus [53]. It is well known that TAMs are abundant within the TME and are recruited to enhance the immunosuppressive environment [54]. Given that high numbers of macrophages are home to tumours, the researchers opted to exploit this and used macrophages to deliver oncolytic viruses to prostate tumours grown in spherical cell complexes that mimic the TME in vitro and in xenograft models of prostate cancer. The macrophage–virus complex successfully delivered the virus to the tumours, resulting in efficient viral replication under hypoxic conditions, tumour oncolysis, and inhibition of tumour growth in mice. More importantly, when co-cultured with high-titre NAb in human serum, the macrophages protected the virus, and was significantly more effective than adenovirus on its own, which was completely neutralised [53]. Additional cellular carriers have also been explored. For example, Melzer et al. showed that CD8+ T central memory cells (CD8+ T cm) can be efficiently loaded with VSV and transport virus to tumour cells without compromising their own viability or antitumor reactivity [55]. Furthermore, mesenchymal stromal cells have been shown to systemically deliver a binary vector containing an OAd together with a helper-dependent Ad (HDAd; combinatorial Ad vector (Cad)) that expresses interleukin-12 (IL-12) and checkpoint PD-L1 (programmed death-ligand 1) blocker [56]. Alternatively, chemical agents can also be used as protective “camouflage” for OV, and this was demonstrated by Nosaki et al. [50]. Ionic polymer coating made via polyethyleneimine hydrochloride was used to encapsulate a measles virus. The coated oncolytic virus was administered to mice bearing LL/2 lung cancer cells. The study demonstrated the enhanced oncolytic activity in the presence of NAb (mice were pre-immunised 3 weeks prior to treatment), with significantly decreased tumour burden. In vitro analysis showed reduced neutralisation of coated MV in multiple cell lines, including MDA-MB-231s (breast), WiDr (colon), and A549 (lung) cells, highlighting the potential for synthetic polymers as effective “shields” against immune elimination [50].

## 8. Overcoming Barriers Using a Targeted Delivery Approach—Magnetic Guided Delivery

Within biomedical applications, synthetic magnetic nanoparticles (MNPs) are easily generated at high yield for a low cost. This, coupled with their strong magnetic properties, makes them ideal candidates as effective drug delivery systems that can be guided via an external magnetic force. The use of magnetic nanocarriers for the delivery of OV was recently reviewed by Howard and Muthana 2020 [51]. So far, this research has mainly used MNPs to investigate the movement of magnetically labelled chemotherapies both in vitro and in vivo or to bind and block viral entry into the cell as an anti-infection treatment [57]. Magnetic guidance could also be used for targeting oncolytic viruses to tumours. Almstatter et al. were the first to publish the in vivo application of MNPs linked to an oncolytic vesicular stomatitis virus (VSV) [58]. The anti-cancer properties of the magnetically labelled virus, along with its MRI contrast properties, were investigated in rats bearing orthotopic hepatocellular carcinoma. Good bioavailability was demonstrated by large aggregates of armed MNPs at the tumour site after visualisation using MRI. Within the experimental process, the tissue-mimicking phantom properties of the MNP–VSV complex indicated the potential for a sustainable contrast agent, which could be visualised for up to 24 h post treatment. In addition, good complex stability was observed in vivo when rats were subjected to an external magnetic gradient (1.5 T and 3 T) for 30 min [58]. Unfortunately, in this study they did not investigate systemic delivery of the magnetised oncolytic virus, as the complex was injected directly into the tumour and the magnetic field was used to keep the complex in the tumour for longer periods. Therefore, it is impossible to predict whether application of magnetic gradients would have improved the targeting of the virus following systemic delivery, or whether the MNPs protected the virus from NAbs.

Tresilwised et al., in an attempt to combat multi-drug resistance (MDR) in cancer, also magnetically labelled an oncolytic adenovirus [59]. As Almstatter et al. [58] did, the particles were intracellularly internalised by 181RDB cells (a MDR pancreatic carcinoma cell line). Ultrastructural analysis showed excellent structural stability and good anti-tumoural killing. Mouse tumour pancreatic xenografts, which were treated IT with the magnetically labelled adenovirus, showed a statistically significant reduction (49%) in tumour burden compared to untreated mice, and in mice treated with adenovirus alone (without a magnetic gradient), 25 days post administration. The magnetically targeted virus was localised via an external static magnetic gradient over the right flank of the mouse to direct the magnetic gradient as much as possible over the pancreas [59]. Similarly to Almstatter et al. [58], no IV administered data of the complex were investigated in this study.

Muthana et al. [60] also investigated whether magnetic guidance could be used to improve targeting of systemically delivered macrophages armed with oncolytic HSV1716 in order to increase tumour specificity. Here the researchers exploited the gradients of MRI scanners to generate a controlled magnetic gradient to non-invasively “steer” the magnetically labelled macrophages from circulation into tumours. This exciting study not only used MRI to guide the therapy to the tumour but to also track delivery using MRI in its conventional imaging modality. They demonstrated a significant increase in drug delivery and reduction in tumour burden in mice with both primary and secondary prostate tumours, after IV administration with the magnetically labelled macrophage–HSV complex compared to virus alone [60].

These studies show that magnetic targeting increases viral titres within the tumours and could be used to enhance the delivery and retention of oncolytic viruses to tumours. The applicability of this approach to breast cancer remains to be investigated.

## 9. Overcoming Breast Cancer-Specific Challenges

Aside from the generic barriers with delivery and enhancing efficacy, breast cancer has some unique challenges. These can be subdivided into host specific (immune system differences in females and the effect of menopause on the immune system) and tumour specific (ER/HER2 status and use of anti-oestrogen therapy). However, research into this interesting area is limited and this section is based on the evidence with immunotherapies in the general cancer population.

Differences in immune response to checkpoint inhibition have been observed between males and females in the treatment of lung cancer [61] and melanoma [62]. This is felt to be because oestrogen acts as a steroid hormone on the immune system, perhaps modulating the tumour environment by promoting a tumourigenic landscape. Oestrogen is thought to play an important role in the adaptive immune system and upregulation of the ER has been observed in T cells and B cells. Furthermore, it may be that disease progression is a consequence of the mobilization of myeloid-deprived suppressor cells and enhancement of their immunosuppressive effects in vivo [63]. Within lung cancer, the use of anti-oestrogens to modulate the immune environment is currently under exploration [64].

Breast cancer is almost exclusively diagnosed in women the vast majority of whom are oestrogen dependent with upregulation of the oestrogen receptor, yet it is in this group that we see the least response to immunotherapies. However, there appears to be a potential role for the use of combination anti-oestrogen therapy, a well-established treatment modality for these cancers and checkpoint inhibition in breast cancer [65].

As a consequence of this, several clinical trials are currently underway with the goal of evaluating the added benefit of oestrogen-modulating drugs to immune checkpoint inhibitors in the context of breast cancer (NCT02997995, NCT02778685, NCT03280563, NCT02990845, NCT02971748, NCT02648477, NCT02971761, NCT02997995 [66]). Within these trials, all combine inhibitors targeting checkpoints CTLA-4, PD-1, or PD-L1 with agents that target the oestrogen pathway, such as fulvestrant or exemestane. Interestingly, oestrogen may also be combined with oncolytic viruses. In a study by Stiles et al., the addition of oestrogen to the oncolytic HSV-1 NV1066 enhanced the lytic effect of the virus, with an MCF-7 cell death of 95% and 97% in vitro at MOIs of 0.1 and 0.5, respectively, compared to 53% and 87%, respectively, in the absence of estrogen [67]. This finding may be used to target the more immunotherapy-resistant ER+ breast cancers and is worth potential investigation.

The other difference between males and females is the predisposition of females to present with autoimmune disease. T helper (Th) cells are postulated to be responsible for this, and females have been found to have a Th1 bias [68]. This may have implications in terms of response to treatment and potential toxicity. Indeed, our own unpublished data have suggested that predisposition to a particular Th cell response generated opposite results. Inoculation with oncolytic HSV virus had a more marked systemic response in tumour-laden Th2-biased Balb/c mice, but a protective one in Th1-biased C57Bl/6 mice (unpublished data).

Breast cancer is also a disease in older women, and there may be an impact of age and response to immunotherapies. Firstly, the changes in an ageing immune system have been extensively covered in the literature [69]. This predisposes the older population to a number of age-related diseases, including a predisposition to Th1-biased diseases such as atherosclerosis and leading to the elderly being poor stimulators of the adaptive immune response and antibody production. In particular, ageing leads to an increased susceptibility to acquiring viral infections and an inadequate immune response [70]. Of interest, ageing leads to an increase in T reg cells, which play a role in masking the cancer from the host’s immune system. This change may be potentially reversed with OV.

Furthermore, hormonal changes associated with menopause have been described as provoking immune-related changes. For example, post-menopausal women show elevated levels of pro-inflammatory cytokines MCP1, TNFalpha, and IL-6 [71]. In many breast cancer patients, menopause is artificially induced with chemotherapy or chemical castration using LHRH antagonists. What is interesting is that in the premature menopause setting, circulating immune cells of the adaptive immune system are modified. Kumru et al. [72] and Giglio et al. [73] analysed the peripheral blood of breast cancer patients who had undergone menopause surgically or naturally, respectively. In both studies a decrease in CD4 T cells and B cells was observed as well as a corresponding increase in CD8 T cells peripherally. These consistent differences raise the question of whether pre- and post-menopausal women respond differently to immunotherapies including OVs, a question that has yet to be addressed in clinical trials.

The next most common subtype of breast cancer is the HER2+ subgroup and accounts for 15–19% of breast cancers [74]. This patient group may respond to immunotherapy and there is some data to suggest that HER2 receptor blockers may cause long-term disease-free survival through this route. To enhance this, there has been interest in developing HER2 vaccinations that consist of HER2 antigens to stimulate an immune response. Morse et al. described a pilot study of one such vaccine administered with dendritic cells to boost immunogenic stimulation [75]. They co-administered dendritic cells (derived from patients’ peripheral blood mononuclear cells) with HER2 antigens and recorded acute and long-term toxicity and response. They describe an impressive 4.5 year survival in all patients, with only one recurring with a single pulmonary lesion at 4 years post treatment. Results from later phase clinical trials have not been reported. Additionally, as HER2 is a targetable receptor, it can be exploited in OV. For example, to specifically target HER2+ breast cancer, the monoclonal antibody trastuzumab is often used clinically to block the ERB receptor. To achieve this, in a preclinical study one group engineered an oncolytic virus where the human trastuzumab antibody heavy- and light-chain genes were uncoded within a serotype 5 adenovirus, Ad5/3-Δ24-tras. This allowed viral oncolysis and assembly and release of the trastuzumab antibody within the TME, with impressive results in vivo [76].

## 10. Conclusions

Breast cancer is a heterogeneous disease and although conventional treatments have evolved significantly over the past few years, treatment resistance invariably occurs. Training the immune system to recognize and target cancers has proved curative for select patients in a number of solid tumour types. The hallmark of response appears to be a favourable “inflammatory” TME. OV, through the production of TAA and immunogenic cell death, may be the platform to sensitise breast cancer to other immunotherapies. However, further work needs to be done to overcome the barriers to OV and personalise treatment for this particular group of patients. This review highlights the areas of current development, which include optimising delivery of oncolytic viruses, combining OV with conventional breast cancer therapeutics, and targeting breast cancer-specific challenges.

## Figures and Tables

**Figure 1 viruses-13-01128-f001:**
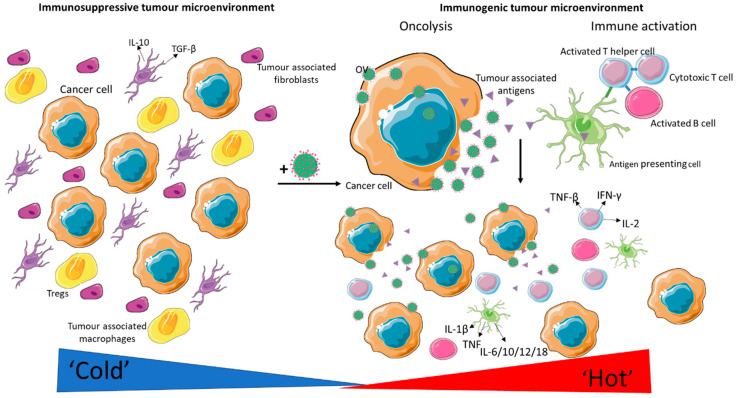
Immunogenic stimulation secondary to oncolytic viruses: Upon viral entry, replication and tumour cell lysis occurs and the innate and adaptive immune systems are activated. The killing of tumour cells via oncolysis releases tumour-associated antigens (TAA) into the circulation. Tumour debris stimulates the activation of resident and circulating antigen-presenting cells, resulting in their maturation. Mature antigen-presenting cells prime both B and T lymphocytes against specific TAAs, leading to long-term immunisation.

**Figure 2 viruses-13-01128-f002:**
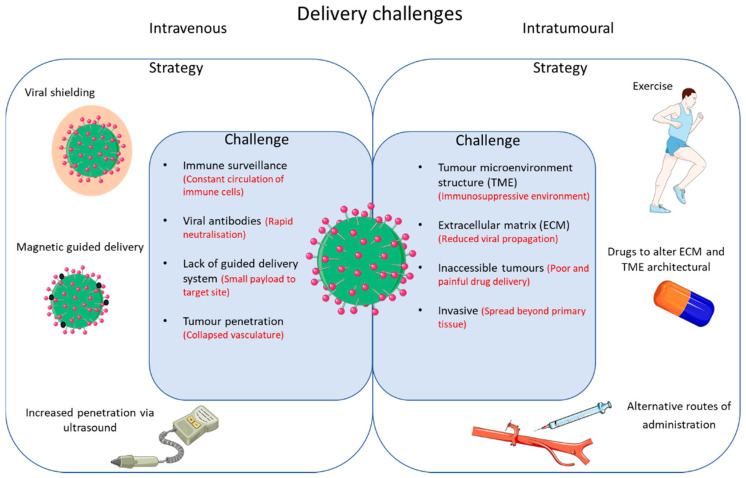
Outlined are some of the challenges and strategies in overcoming restricted delivery of OVs.

**Table 1 viruses-13-01128-t001:** Combination of oncolytic viruses and checkpoint inhibition in breast cancer.

OV	Checkpoint Inhibitor	Cancer	Reference
Oncolytic vaccinia virus co-expressing a mouse PD-L1 inhibitor and GM-CSF.	PD-L1 inhibitor	Py230 breast cancer	[43]
Oncolytic reovirus—non-modified	PD-1 inhibitor	Immunocompetent, syngeneic EMT6	[30]
Polio:rhinovirus recombinant (PVSRIPO)	PD1/PD-L1 axis	E0771	[44]
Marabavirus—non-modified	Anti-PD-1 (clone RMPI-14, BioXCell) and anti-CTLA4 (clone 9D9, BioXCell)	EMT6, E0771, 4T1 immunocompetent, syngeneic neoadjuvant models	[41]
Modified measles virus (MV-NAP)	PD-1/PD-L1 blockade	Phase 1 trial ongoing (Mayo Clinic Breast Cancer SPORE)	https://grantome.com/grant/NIH/P50-CA116201-12 (accessed on: 1 May 2021)

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
