# Peer review of "Oncolytic Virotherapy Treatment of Breast Cancer: Barriers and Recent Advances"

_viruses, 2021, doi:10.3390/v13061128_

Round 1

Reviewer 1 Report

The authors conducted a comprehensive review of oncolytic virotherapy for breast cancer.

This review, which focuses on the combination of OV and other therapies, barriers to OV, and overcoming barriers, is valuable in the development of oncolytic virotherapy for breast cancer.

However, there are some points to reconsider, as described below.

Page 1, line 44: “HER2+ receptor” would be “human epidermal growth factor receptor 2 (HER2)+”.

”Page 3, Figure 1: More detailed explanations are required in figure legend. Do cytotoxic T cells destroy cancer cells? How do activated B cells function in this situation? “T reg” can be “Treg”. “TNFb” and “IFNg” can be “TNF-b” and “IFN-g”, respectively.

Page 3, line 120: “T regs” can be “Tregs”.

Page 4, line 129: Does “the addition of a GM-CSF promoter” mean “an oncolytic adenovirus coding for GM-CSF was engineered”?

Page 4, lines 152-156: In reference [24], patients were given HF10 to metastatic nodules, whereas in reference [25], G47 delta was given intravenously to nude mice with tumors. Since the background pf the experiment is different between these two studies, it is difficult to discuss the systemic immune response from the same point of view.

Page 6, line 227: A phase II trial of paclitaxel and pelareorep in patients with metastatic breast cancer was reported by Bernstein et al (Breast Cancer Res Treat 2018, 167: 485). This can be added to the reference list.

Page 6, line 265 and page 7, lines 267, 269: “TVEC” becomes “T-Vec”.

Page 9, line 376: Other systems have been developed to deliver oncolytic viruses. These can be added in the text and reference.

   Melzer-MK, et al. Enhanced safety and efficacy of oncolytic VSV therapy by combination with T cell receptor transgenic T cells as carriers. Mol Ther Oncolytics 2018; 12: 26.

   McKenna-MK, et al. Mesenchymal stromal cell delivery of oncolytic immunotherapy improves CAR-T cell antitumor activity. Mol Ther. 2021; 29: 1808.

Author Response

Dear Reviewer 1, 

Thank you very much for taking the time to review our manuscript. Your comments and insight are extremely valuable and we have amended the manuscript accordingly. We hope the amended version is to your satisfaction. 

Kind regards

Dr Amy Kwan 

Reviewer 2 Report

This review article discusses oncolytic virotherapy approaches in breast cancer. The authors detail current approaches using monotherapy and oncolytic viruses combined with others immunotherapies, and traditional approaches including chemotherapy and radiotherapy. Overall, this article is well-written and well-detailed. I have a couple minor comments and suggestions before publication:

1) The abbreviations on page 3 should be moved to the beginning or the end of the article.

2) Though not specific to breast cancer, the authors may be interested in the following papers. The review by Martin and Bell that discusses combination OV approaches (https://doi.org/10.1016/j.ymthe.2018.04.001), specifically the section on combining different viruses. Combining different viruses has been studied in particular by Le Boeuf et al. (https://doi.org/10.1038/mt.2010.44) in HT29 cell lines. Using a computational model, Jenner et al. (http://dx.doi.org/10.1136/jitc-2020-001387) looked at the determinants of combination OV therapeutic success and found it linked to tumour aggressivity (model calibrated to HT29 and mammary carcinoma cell lines).

Author Response

Thank you very much for taking the time to review our manuscript. Your comments and insight are extremely valuable and we have amended the manuscript accordingly. We hope the amended version is to your satisfaction. 

Kind regards

Dr Amy Kwan